

# The force-length relation of the young adult human tibialis anterior

Brent J. Raiteri[1], Leon Lauret[1] and Daniel Hahn[1,2]

[1] Human Movement Science, Faculty of Sport Science, Ruhr-Universität Bochum, Bochum, Nordrhein-Westfalen, Germany
[2] School of Human Movement and Nutrition Sciences, University of Queensland, Brisbane, Queensland, Australia

## ABSTRACT

**Background**. Knowledge of the muscle's lengths at which maximum active isometric force is attained is important for predicting forces during movement. However, there is limited information about the *in vivo* force-length properties of a human muscle that plays crucial roles during locomotion; the tibialis anterior (TA). We therefore aimed to estimate TA's force-length relation from dorsiflexor torque-angle curves constructed from eight women and eight men.

**Methods**. Participants performed maximal voluntary fixed-end contractions with their right ankle dorsiflexors from 0° to 30° plantar flexion. Muscle fascicle lengths were estimated from B-mode ultrasound images, and net ankle joint torques were measured using dynamometry. Fascicle forces were estimated by dividing maximal active torques by literature-derived, angle-specific tendon moment arm lengths while assuming a fixed 50% force contribution of TA to the total dorsiflexor force and accounting for fascicle angles.

**Results**. Maximal active torques were higher at 15° than 20° and 30° plantar flexion ($2.4$–$6.4$ Nm, $p \leq 0.012$), whereas maximal active TA fascicle forces were higher at 15° than 0°, 20° and 30° plantar flexion ($25$–$61$ N, $p \leq 0.042$), but not different between 15° and 10° plantar flexion ($15$ N, $p = 0.277$). TA fascicle shortening magnitudes during fixed-end contractions were larger at 15° than 30° plantar flexion ($3.9$ mm, $p = 0.012$), but less at 15° than 0° plantar flexion ($-2.4$ mm, $p = 0.001$), with no significant differences ($\leq 0.7$ mm, $p = 0.871$) between TA's superficial and deep muscle compartments. Series elastic element stiffness was lowest and highest at lengths 5% shorter and 5% longer than optimum fascicle length, respectively ($-30$ and $15$ N/mm, $p \leq 0.003$).

**Discussion**. TA produced its maximum active force at 10–15° plantar flexion, and its normalized force-length relation had ascending and descending limbs that agreed with a simple scaled sarcomere model when active fascicle lengths from within TA's superficial or deep muscle compartment were considered. These findings can be used to inform the properties of the contractile and series elastic elements of Hill-type muscle models.

Corresponding author
Brent J. Raiteri,
brent.raiteri@gmail.com

# INTRODUCTION

It would be a trivial task to understand how muscles function to power movement if muscle forces could be directly and non-invasively measured. To date, measuring *in vivo* muscle force is not possible, so muscle force is commonly predicted through mechanical models (*Hill, 1938*; *Huxley, 1957*). Muscle models are used in simulations of human movement to understand muscle function (*Hamner, Seth & Delp, 2010*) and motor control (*Berniker et al., 2009*), as well as estimate joint loading (*Curreli et al., 2021*), and plan surgery (*Fox et al., 2009*). Accurate estimation of muscle forces in clinical contexts is of particular importance, but the properties of the underlying muscle models are not typically based on *in vivo* human data. This is a major limitation because the force output from such models is very sensitive to the contractile element's force-length curve (*Scovil & Ronsky, 2006*), and curves likely differ between artificially-activated isolated animal muscles and voluntarily-activated intact human muscles (*Herzog & ter Keurs, 1988*).

Despite the importance of collecting human skeletal muscle force-length data to inform muscle models and subsequently improve the accuracy and performance of musculoskeletal simulations (*Scovil & Ronsky, 2006*), only few studies have made *in vivo* estimates, and these estimates typically come from healthy, young men (*Herzog, Read & Ter Keurs, 1991*; *Herzog & ter Keurs, 1988*; *Maganaris, 2001*; *Oda et al., 2005*). This lack of information from women is understandable given the sex bias in neuroscience and biomedical research (*Beery & Zucker, 2011*), and the general lack of data from intact human muscles is primarily because of the difficulties associated with estimating individual muscle forces from net joint torque measurements. Multiple muscle–tendon units cross the same joint *in vivo*, which results in the force-length relations of multiple muscles contributing to a joint's torque–angle relation. Moreover, these muscle–tendon units have unique moment arms that are affected by the joint's configuration (*Murray, Buchanan & Delp, 2002*), and typically increase with increasing force production (*Maganaris, Baltzopoulos & Sargeant, 1999*). As a result, the moment arm-joint angle relation of a muscle–tendon unit of interest should be known before its muscle's force-length relation is estimated (*Rassier, MacIntosh & Herzog, 1999*).

Another measurement that is required to help accurately estimate a muscle's *in vivo* force-length relation is muscle fascicle length. Muscle fascicle length should be quantified while the muscle is active because muscle–tendon unit compliance, which is relatively high within lower limb muscle–tendon units, permits muscle fiber shortening during active force production even when muscle–tendon unit length remains relatively constant (*Muhl, 1982*). Additionally, the amount of shortening for a given muscle–tendon unit force is not necessarily constant at different muscle–tendon unit lengths (*Ichinose et al., 1997*; *Moo, Leonard & Herzog, 2020*; *Raiteri, Cresswell & Lichtwark, 2018*). Consequently, predicting fascicle lengths based on a constant in-series compliance might be flawed, which could help to explain the disagreement between estimated *in vivo* and theoretically-predicted active force-length relations of the human rectus femoris muscle (*Herzog & ter Keurs, 1988*).

Our main aim was to estimate the force-length properties of the human tibialis anterior (TA) from healthy, young women and men during maximal voluntary contractions. This is because only two studies have previously investigated this relation in limited samples of six healthy men *via* supramaximal stimulations of TA's muscle belly and the common peroneal nerve (*Maganaris, 2001*; *Oda et al., 2005*). These non-invasive stimulation methods are problematic, especially at more dorsiflexed angles, because they can activate the peroneus longus and brevis muscles, which oppose the dorsiflexion torque generated by the TA (*Gandevia & McKenzie, 1988*). Consequently, dorsiflexion torque may fail to increase as stimulation intensity increases due to increasing plantar flexion torque contributions. Both studies also did not assess the agreement between experimentally-estimated and theoretically-predicted active force-length relations, or compare differences in muscle architecture between TA's superficial and deep muscle compartments; which has only been done once before, again in a limited sample of six healthy men (*Maganaris & Baltzopoulos, 1999*). Therefore, an *in vivo* study on TA's force-length relation under maximal voluntary conditions on a larger sample including women appears warranted to help interpret the relative force-producing capacity of TA during everyday movements, such as walking, and to inform the contractile and series elastic element properties of Hill-type muscle models used in human movement simulations.

We hypothesized that there would be optimum angles of dorsiflexion torque and TA fascicle force production based on previously reported dorsiflexor torque–angle (*Marsh et al., 1981*) and TA force-length (*Oda et al., 2005*) relations. We also hypothesized that TA's muscle fascicle shortening magnitudes during fixed-end contractions would be significantly larger at shorter muscle–tendon unit lengths based on previous work (*Raiteri, Cresswell & Lichtwark, 2018*), which would result in a decreasing series elastic element stiffness with decreasing muscle–tendon unit length. Our third hypothesis was that fascicle shortening and work magnitudes would not be significantly different between muscle compartments because of their common distal insertion and similar architecture (*Maganaris & Baltzopoulos, 1999*). Our fourth and final hypothesis was that normalized force-length relations based on active and passive fascicle lengths would poorly match theoretical estimates (*i.e.,* root-mean-squared errors of over 10%) based on human actin and myosin filament lengths, which is in line with previous *in vivo* work on the human rectus femoris (*Herzog & ter Keurs, 1988*).

## MATERIALS & METHODS

### Participants

Sixteen healthy participants (eight women and eight men) with no recent history (<24 months) of surgery, major injury, or minor (<6 months) injury to the lower limb participated in the study after providing written informed consent. All participants were physically active and occasionally or frequently took part in training for recreation or sport, but no participants had previously trained their dorsiflexor muscles. Participants had a mean (standard deviation) age of 26.3 (2.6) years (range: 23–33 years), height of 177.3 (9.6) cm (range: 162–196 cm), and weight of 72.2 (12.8) kg (range: 56–98 kg).

The experimental protocol was approved by the Ethics Committee of the Faculty of Sport Science at Ruhr University Bochum (EKS V 33/2019). The study was conducted in accordance with the ethical principles outlined within the Declaration of Helsinki with the exception that the study was not pre-registered.

## Experimental set-up

The experimental setup is illustrated in Fig. 1 and further details are provided in the caption. Participants performed maximal voluntary fixed-end dorsiflexion contractions with their right dorsiflexors while they sat in a reclined posture (∼115° backrest relative to seat) on the seat of a motorized dynamometer. Supramaximal stimulations of the common or deep peroneal nerve or TA muscle belly were not delivered during maximal voluntary contractions to assess voluntary activation because of the problems associated with co-activating the peroneus longus and brevis muscles, which oppose the dorsiflexion torque generated by the TA, especially at more dorsiflexed angles (*Gandevia & McKenzie, 1988*). However, previous work suggests that the TA can also be maximally voluntarily activated in dorsiflexed and plantar flexed positions as assessed *via* intramuscular stimulation of its proximal motor point (*Gandevia & McKenzie, 1988*), so we assumed this to be the case during our experiments. The plantar aspect of the right foot was secured to the footplate of a dorsi/plantar flexion adapter by tightening a custom-made dorsiflexion attachment (Article S1) over the superior aspects of the metatarsals. The right thigh and foot were vertically aligned in the frontal plane, and the right knee and hip were flexed to approximately 90° and 110° in the sagittal plane, respectively (Fig. 1).

A 6 cm flat linear-array ultrasound transducer (LV8-5N60-A2, TELEMED, Vilnius, Lithuania) was secured with a constant pressure over the mid-belly of the right TA muscle using self-adhesive bandage (7.5 cm width, COJJ, Amazon, Seattle, Washington, United States) to image muscle fascicles within TA's superficial and deep compartments (*Raiteri, Cresswell & Lichtwark, 2016a*) at ∼137 fps using a coupled PC-based ultrasound system (ArtUs EXT-1H, TELEMED). Two surface electrodes (hydrogel Ag/AgCl, 8 mm recording diameter, H124SG, Kendall International Inc., Mansfield, Massachusetts, United States) were taped (1.25 cm width surgical tape, 3M Transpore, St. Paul, Minnesota, United States) to the skin in a bipolar configuration immediately distal to the ultrasound transducer and above TA's superficial compartment, and a single reference electrode was taped over the right fibular head, to record the myoelectric signal over TA at 2 kHz using surface electromyography (NL905, Digitimer Ltd.; Welwyn Garden City, United Kingdom). Further details about these techniques are provided in Article S2.

A neutral ankle joint angle was defined as a 90° angle between the tibia and footplate of the dynamometer adapter and will be termed 0° plantar flexion from here on. This angle was controlled by ensuring the superior aspects of the tibia and footplate were at 45 ± 1° relative to level ground as measured by a digital inclinometer (± 0.1° accuracy; Beaspire; Amazon, Seattle, WA, United States). The ankle joint's axis of rotation and dynamometer's axis of rotation were aligned at 15° plantar flexion during a maximal voluntary fixed-end dorsiflexion contraction with the help of a laser pointer. The laser pointer projected the dynamometer's axis of rotation onto the skin, and the ankle joint's axis of rotation was

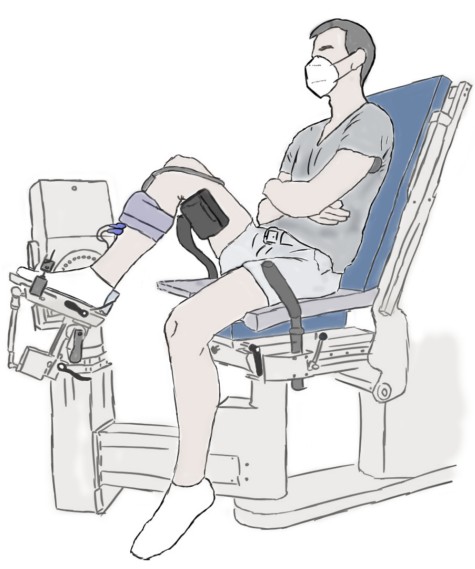

**Figure 1 Illustration of the experimental setup.** A dorsiflexion attachment over the metatarsals limited ankle joint rotation and force contributions from the toe extensors to the measured net ankle joint torque during maximal voluntary fixed-end dorsiflexion contractions. A cushioned support was fixed underneath the distal right femur to support the right thigh. A 6 cm flat linear-array ultrasound transducer was located underneath the self-adhesive strap around the right mid-shank to image the tibialis anterior (TA) muscle with B-mode ultrasound. The two surface electrodes immediately distal to the transducer recorded the single differential myoelectric signal over the TA *via* surface electromyography. Participants were instructed to fold their arms across their stomach and to focus on pulling the top of their foot towards their shank using only their dorsiflexor muscles.

considered aligned when the laser was located over the palpated lateral malleolus and in line with an imaginary vector passing through the most prominent aspects of the medial and lateral malleoli. The footplate was inverted by 5° about its longitudinal axis to ensure that the dynamometer and ankle joint axes of rotation were approximately parallel.

## Experimental protocol

Participants took part in one experimental session and first performed five submaximal (1-s hold, 1-s rest, ∼80% of perceived maximum) voluntary dorsiflexion contractions at 15° plantar flexion to pre-condition the dorsiflexor muscle–tendon units (*Maganaris, Baltzopoulos & Sargeant, 2002*). Following a two-minute break, participants then performed one to five maximal (3-s to 5-s hold) voluntary dorsiflexion contractions at a minimum of five plantar flexion angles in a randomized order, which included 0°, 10°, 15°, 20°, and 30° plantar flexion. Additional maximal voluntary dorsiflexion contractions were performed at −15°, −10°, −5°, 5°, and 40° plantar flexion if this was required to construct a dorsiflexor torque–angle relation with clear ascending and descending limbs. The maximal voluntary contraction that exhibited the highest peak torque was always repeated and in half of the participants, maximal voluntary contractions were repeated at one ($n = 3$), two ($n = 3$), or three ($n = 2$) additional plantar flexion angles. Standardized verbal encouragement was provided by the investigator during each contraction and real-time visual feedback of the recorded net ankle joint torque was shown to the participants

*via* a screen positioned in front of them. At least two minutes were provided between contractions to minimize fatigue. Contractions were not repeated at all tested plantar flexion angles to avoid fatigue in a subsequent part of the experiment designed to investigate another research aim. At the end of the experiment, six passive ankle rotations were performed at $5° \text{ s}^{-1}$ over each participant's joint range of motion.

## Dynamometry

A motorized dynamometer (IsoMed2000; D&R Ferstl GmbH, Hemau, Germany) measured net ankle joint torque and the angle of the footplate relative to the fixed right shank during voluntary dorsiflexion contractions. The dynamometer was equipped with a KISTLER torque sensor (Kistler, Sindelfingen, Germany), which had an uncertainty of <0.1% for a 750 Nm range. Torque and angle data were sampled at 2 kHz and synchronized using a 16-bit analog-to-digital converter (Power1401-3) and Spike2 (8.23 64-bit version) data collection system (Cambridge Electronic Design Ltd., Cambridge, United Kingdom), which was set to a $\pm$ 5 V input range. This data collection system was used to record all digital signals.

## Data processing and analysis

The recorded digital signals from each trial were first combined with the tracked ultrasound data using custom-written scripts in Spike2 and MATLAB (Article S3). Absolute muscle fascicle lengths and fascicle angles (relative to the horizontal) of one representative fascicle from each of the superficial and deep muscle compartments of TA were included in the tracked ultrasound data and were calculated in MATLAB (Fig. 2) using an updated version of UltraTrack (*Farris & Lichtwark, 2016*). Further details about the absolute fascicle length determinations and length change estimates are provided in Article S4.

To synchronize all data, a time vector for each trial was constructed using the timestamps of the torque signal and each signal was subsequently resampled using linear interpolation. Torque and angle data were filtered using zero-lag second-order 20 Hz and 6 Hz low-pass Butterworth filters, respectively, which were corrected for two passes (*Winter, 2009*). To calculate active torque, a steady-state passive ankle joint torque–angle fit was constructed, then evaluated at the plantar flexion angles recorded during each trial and subtracted from the recorded net ankle joint torque (Article S5).

Maximal active joint torque and the corresponding plantar flexion angle were determined for each trial, and the trial with the highest active joint torque was analyzed when multiple contractions were performed at the same plantar flexion angle. Active dorsiflexor force ($F_{\text{active}}$) was estimated by the following equation:

$$F_{\text{active}}(N) = \frac{T_{\text{active}}(Nm)}{r_{\text{active}}(m)}$$

where $T_{\text{active}}$ represents the maximal active torque, and $r_{\text{active}}$ represents a literature-derived, joint-angle-specific, MRI-based muscle–tendon moment-arm length during maximal voluntary contraction (*Maganaris, Baltzopoulos & Sargeant, 1999*). TA tendon force ($F_{\text{tendon}}$) was calculated by:

$$F_{\text{tendon}}(N) = F_{\text{active}}(N) \times 0.5$$

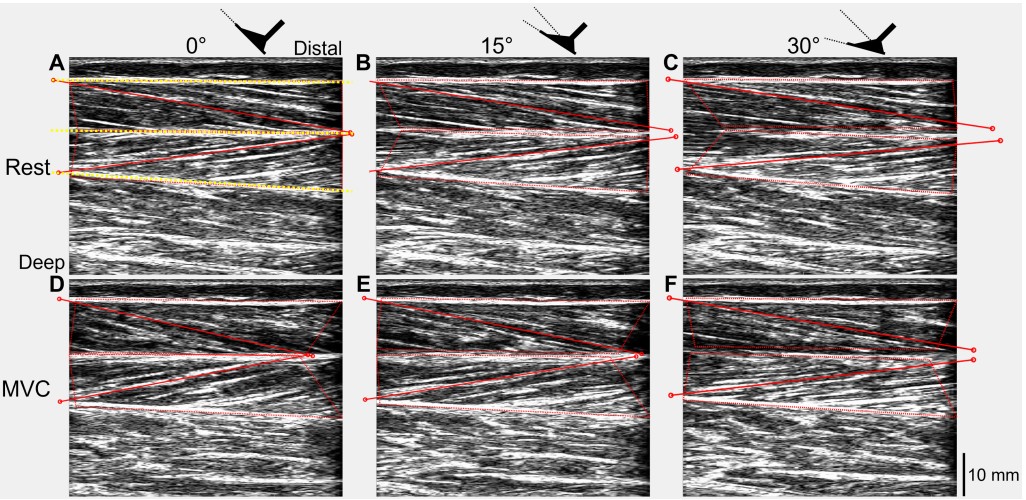

**Figure 2** **Ultrasound images of a participant's tibialis anterior muscle at rest (A–C) and during maximal voluntary dorsiflexion contractions (D–F) at 0° (A&D), 15° (B&E), and 30° (C&F) plantar flexion.** One representative fascicle (and its endpoints) from each of tibialis anterior's (TA's) superficial and deep muscle compartments are indicated by a red solid line (and red unfilled circles) in each image. TA's superficial, central, and deep aponeuroses are indicated by the yellow dotted lines in A.

as TA was assumed to contribute 50% of the total dorsiflexor force at all tested plantar flexion angles (*Brand, Pedersen & Friederich, 1986*; *De Zee & Voigt, 2002*; *Maganaris & Paul, 2000*). TA fascicle force ($F_{fascicle}$) was calculated as:

$$F_{fascicle}(N) = \frac{F_{tendon}(N)}{\cos\theta(°)}$$

where $\theta$ represents the respective fascicle angle.

TA fascicle work was calculated as the area under the estimated fascicle force–displacement curve using the trapezoidal method. This curve started at the instant of fascicle shortening (*i.e.,* zero displacement) and ended at the shortest fascicle length (*i.e.,* maximum displacement), and the fascicle shortening magnitude within each muscle compartment was calculated as the difference between these two values. The instant of fascicle shortening was determined by finding the first abrupt change in the slope and intercept of the fascicle length data using a piece-wise linear regression. The maximal active force and corresponding plantar flexion angle were determined for each trial at the shortest fascicle length. Series elastic element stiffness was calculated as the slope of the relation between horizontal fascicle displacement and TA tendon force between 40% and 90% of maximum force. Mean series elastic element stiffness at lengths: (1) 5% shorter than optimum fascicle length (*i.e.,* ascending limb of the force-length relation); (2) 5% longer than optimum fascicle length (*i.e.,* descending limb), and; (3) between (1) and (2) (*i.e.,* plateau region) was then taken. Horizontal fascicle displacement, which represents the length change of the series elastic element ($\Delta L_{SEE}$), was determined at each instant in time by:

$$\Delta L_{SEE}(mm) = \Delta L_{fascicle}(mm) \times \cos\theta(°)$$

where $\Delta L_{\text{fascicle}}$ represents fascicle displacement.

## Statistics

Statistical analysis was performed with GraphPad Prism software (9.1.2 64-bit version, San Diego, California, USA) and the alpha level was set at 5%. TA fascicle force and work were not estimated from two participants (one woman and one man) because the ultrasound video durations were not equal to the analog data durations ($\geq 0.03$ s). A one-way repeated-measures mixed-effects analysis with the Greenhouse–Geisser correction was performed to identify differences in maximal active dorsiflexion torques across plantar flexion angles of 0°, 10°, 15°, 20°, and 30° ($n = 12$ as four participants did not perform maximal voluntary efforts at 0° plantar flexion). The same analysis was performed to identify differences in series elastic element stiffness across different regions of TA's force-length relation. Two-way repeated-measures mixed-effects analyses with the Geisser-Greenhouse correction were performed to identify differences in maximal fascicle forces and fascicle shortening and work amplitudes between muscle compartments across the plantar flexion angles mentioned above (muscle compartment × plantar flexion angle). Following a significant plantar flexion angle effect or interaction, individual-variance-based Holm-Sidak multiple comparisons were performed between each angle and the mid-range angle (*i.e.,* 15° plantar flexion).

Intra-session test-retest reliabilities of fascicle shortening and work amplitudes were calculated from a subset of participants with ultrasound data who attained maximal torques within 15% of each other ($n = 13$) during repeated maximal voluntary dorsiflexion contractions at their angle of maximum torque production. The intraclass correlation coefficient ($ICC_{3,1}$) and 95% confidence interval (95% CI) were estimated in MATLAB using a single-measurement, absolute-agreement, two-way mixed-effects model (*Koo & Li, 2016*). The standard error of measurement (*SEM*) was estimated as the square root of the mean square error term from a one-way repeated-measures ANOVA (*Weir, 2005*). The minimum detectable change ($MDC_{95}$) was estimated by:

$$MDC_{95} = SEM \times \sqrt{2} \times t$$

where $t$ refers to the 95% confidence interval critical value from a two-tailed $t$-distribution with $n - 1$ degrees-of-freedom (*Weir, 2005*). Data are reported as mean (standard deviation, SD) in the text and figures.

## RESULTS

### Dorsiflexor torque–angle relation

Maximal active dorsiflexion torques were significantly different across plantar flexion angles ($F_{1.39,15.24} = 10.55$, $p = 0.003$) with a higher active torque at 15° plantar flexion than at 20° and 30° plantar flexion (mean differences: 2.4 and 6.4 Nm, respectively, $p \leq 0.012$). Active torques were similar between 15° plantar flexion and 0° or 10° plantar flexion (mean differences: 0.5 and −0.1 Nm, respectively, $p = 0.942$). The mean active dorsiflexor torque–angle relation is shown in Fig. 3A, and descriptive statistics are reported in Table 1.

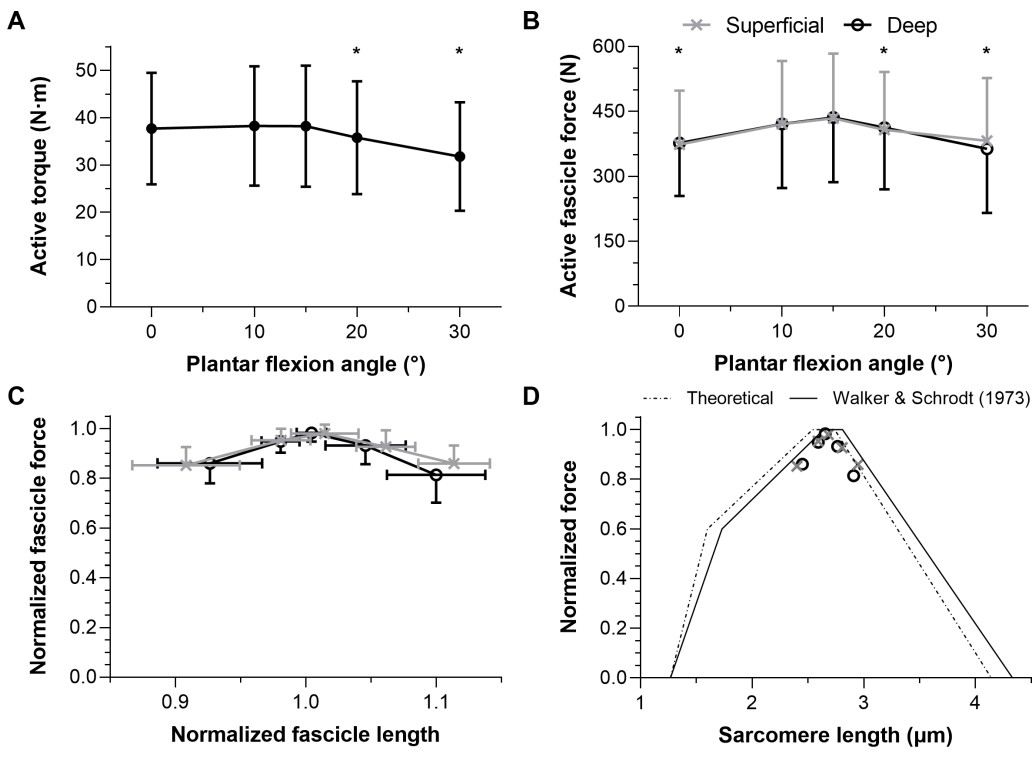

**Figure 3** The mean (SD) maximal active dorsiflexor torque–angle relation (A) and estimated tibialis anterior active force–angle relation (B), and normalized (C) and theoretical (D) force-length relations. Asterisks in A ($n = 12$) and B ($n = 10$) indicate that the mean torque or force was significantly ($p < 0.05$; see results text) lower than the mean torque or force at 15° plantar flexion. Maximum active force and the corresponding fascicle length (*i.e.,* optimum fascicle length) from tibialis anterior's (TA's) superficial or deep compartment were used to normalize individual data before the between-subject mean was taken in C ($n = 10$) and D ($n = 10$). Sarcomere length predictions (D) were calculated by dividing optimum fascicle length by an optimum sarcomere length of 2.64 μm, and then active fascicle lengths at each tested angle were divided by the corresponding sarcomere number. The theoretical human sarcomere length predictions (D) were based on numbers reported in Fig. 13B of *Walker & Schrodt (1974)*, and geometrical predictions that neglect Z-disc width, assume a thick filament bare zone width of 0.2 μm, and human thick and thin filament lengths of 1.60 μm and 1.27 μm, respectively (*Burkholder & Lieber, 2001*).

**Table 1** Mean (SD) maximal active dorsiflexion torques produced at ankle angles from 0° to 30° plantar flexion.

| Ankle angle (°) | 0 | 10 | 15 | 20 | 30 |
|---|---|---|---|---|---|
| *n* of participants | | | 12 | | |
| Torque (N · m) | 37.7 (11.8) | 38.3 (12.6) | 38.2 (12.8) | 35.8 (11.9) | 31.8 (11.5) |

**Notes.**
*n*, number;  N · m,  Newton meter.

## Tibialis anterior force–angle relation

Maximal active TA fascicle forces were significantly different across plantar flexion angles ($F_{1.43,12.88} = 5.53$, $p = 0.026$), but not significantly different between muscle compartments ($F_{1,9} = 0.20$, $p = 0.665$, mean differences: −5 to 18 N), and there was no significant interaction between plantar flexion angle and muscle compartment ($F_{1.50,13.54} = 3.30$,

**Table 2  Mean tibialis anterior moment arm lengths used to estimate mean (SD) maximal active fascicle forces, and normalized fascicle and sarcomere lengths, and work amplitudes from 0° to 30° plantar flexion.** Sarcomere lengths were scaled based on an optimum sarcomere length of 2.64 μm. The force, length, or work amplitude value at the instant of maximum active force production was used to normalize individual data before the between-subject mean was taken.

| Ankle angle (°) | 0 | 10 | 15 | 20 | 30 |
|---|---|---|---|---|---|
| $n$ of participants | | | 10 | | |
| MAL (cm) | 4.9 | 4.4 | 4.2 | 4.1 | 4.0 |
| Muscle fascicle force (N) | 374 (124) | 421 (146) | 434 (150) | 408 (134) | 382 (145) |
| Force norm. | 0.85 (0.07) | 0.95 (0.05) | 0.98 (0.04) | 0.93 (0.07) | 0.86 (0.07) |
| Active FL norm. | 0.91 (0.04) | 0.98 (0.02) | 1.01 (0.03) | 1.06 (0.02) | 1.11 (0.03) |
| Active SL (μm) | 2.40 (0.10) | 2.59 (0.06) | 2.68 (0.06) | 2.81 (0.05) | 2.95 (0.07) |
| Work norm. | 0.81 (0.14) | 0.89 (0.10) | 0.92 (0.12) | 0.75 (0.19) | 0.60 (0.16) |

**Notes.**

$n$, number; MAL, Moment Arm Length; norm., normalized; FL, Fascicle Length; SL, Sarcomere Length.

$p = 0.079$). There was a higher active force at 15° plantar flexion than at 0°, 20° or 30° plantar flexion (mean differences: 61, 25, and 53 N, respectively, $p \leq 0.042$), whereas forces were similar at 15° plantar flexion and 10° plantar flexion (mean difference: 15 N, $p = 0.277$). The estimated mean active TA force–angle relation is shown in Fig. 3B, and descriptive statistics are reported in Table 2.

## Muscle compartment differences

Mean muscle fascicle lengths, fascicle shortening magnitudes and normalized fascicle work amplitudes calculated from both compartments are presented in Table 3. Fascicle shortening magnitudes were significantly different across plantar flexion angles ($F_{1.53,13.78} = 23.46$, $p < 0.001$), but not significantly different between muscle compartments ($F_{1,9} = 0.03$, $p = 0.871$, mean differences: −0.4 to 0.7 mm), and there was no significant interaction between plantar flexion angle and muscle compartment ($F_{1.89,17.02} = 1.53$, $p = 0.244$). There was less fascicle shortening at 15° plantar flexion than at 0° plantar flexion (mean difference: −2.4 mm, $p = 0.001$), despite a significantly higher mean force at 15° plantar flexion. Fascicle shortening magnitudes were similar between 15° plantar flexion and 10° or 20° plantar flexion (mean differences: −0.4 and 2.1 mm, respectively, $p \geq 0.070$), but there was more fascicle shortening at 15° plantar flexion than at 30° plantar flexion (mean difference: 3.9 mm, $p = 0.012$).

Fascicle work magnitudes were also significantly different across plantar flexion angles ($F_{1.69,15.23} = 6.79$, $p = 0.010$), but not significantly different between muscle compartments ($F_{1,9} = 0.20$, $p = 0.669$), and there was no significant interaction between plantar flexion angle and muscle compartment ($F_{1.35,12.18} = 0.86$, $p = 0.406$). There was more positive fascicle work at 15° plantar flexion than at 30° plantar flexion (mean difference: 1.0 J, $p = 0.023$), whereas positive fascicle work magnitudes were similar between 15° plantar flexion and 0°, 10°, or 20° plantar flexion (mean differences: 0.4, 0.2, and 0.7 J, respectively, $p \geq 0.090$).

The intra-session test-retest reliabilities of TA's fascicle shortening and work amplitudes ranged from poor to excellent, and relative reliabilities were superior from the

**Table 3 Mean (SD) tibialis anterior active muscle fascicle lengths, fascicle shortening magnitudes and normalized fascicle shortening work amplitudes from 0° to 30° plantar flexion.** The work amplitude value at the instant of maximum active force production was used to normalize individual data before the between-subject mean was taken.

| Ankle angle (°) | 0 | 10 | 15 | 20 | 30 |
|---|---|---|---|---|---|
| n of participants | | | 10 | | |
| FL sup. (mm) | 63.1 (11.2) | 68.1 (12.0) | 70.4 (11.3) | 73.8 (12.9) | 77.5 (13.4) |
| FL deep (mm) | 73.3 (14.4) | 77.6 (14.1) | 79.4 (14.2) | 82.9 (16.3) | 87.1 (16.5) |
| Shortening amp. sup. (mm) | 14.2 (3.8) | 11.9 (3.2) | 11.2 (3.7) | 9.2 (2.7) | 7.9 (2.1) |
| Shortening amp. deep (mm) | 13.5 (3.8) | 11.8 (3.5) | 11.6 (3.6) | 9.5 (2.4) | 7.3 (2.4) |
| Work amp. norm. sup. (J) | 0.81 (0.13) | 0.89 (0.09) | 0.92 (0.11) | 0.75 (0.18) | 0.60 (0.15) |
| Work amp. norm. deep (J) | 0.78 (0.16) | 0.87 (0.07) | 0.94 (0.08) | 0.76 (0.18) | 0.56 (0.15) |

Notes.

$n$, number; FL, Fascicle Length; sup., superficial; amp., amplitude; norm., normalized.

**Table 4 Intra-session test-retest reliabilities of tibialis anterior's estimated maximum active fascicle forces, corresponding active fascicle lengths, and fascicle shortening and work amplitudes.**

| Reliability metric ($n = 13$) | $ICC_{3,1}$ [95% CI] | SEM | $MDC_{95}$ |
|---|---|---|---|
| Maximum force sup. (N) | 0.99 [0.96 to 1] | 16.7 | 51.4 |
| Maximum force deep (N) | 0.99 [0.96 to 1] | 15.9 | 48.9 |
| Optimum length sup. (mm) | 0.98 [0.93 to 0.99] | 1.5 | 4.8 |
| Optimum length deep (mm) | 0.99 [0.96 to 1] | 1.5 | 4.7 |
| Shortening superficial (mm) | 0.86 [0.60 to 0.95] | 1.5 | 4.5 |
| Shortening deep (mm) | 0.76 [0.38 to 0.92] | 1.4 | 4.4 |
| Work superficial (J) | 0.97 [0.89 to 0.99] | 0.2 | 0.7 |
| Work deep (J) | 0.95 [0.84 to 0.98] | 0.2 | 0.8 |

Notes.

$n$, number; ICC, Intraclass Correlation Coefficient; 95% CI, 95% Confidence Interval; SEM, Standard Error of Measurement; $MDC_{95}$, Minimum Detectable Change based on a 95% CI; sup., superficial.

superficial compartment (Table 4), which is the fascicle plane the ultrasound image plane was aligned to. Consequently, mean normalized fascicle shortening magnitudes from TA's superficial compartment at each plantar flexion angle are shown in Fig. 4A.

## Series elastic element stiffness

Series elastic element stiffness was significantly different over the different regions of TA's force-length relation ($F_{1.38,12.41} = 23.82$, $p < 0.001$). Series elastic element stiffness was lowest at lengths 5% shorter than optimum fascicle length (mean differences relative to the optimum and longer than optimum lengths: $-30$ and $-45$ N/mm, respectively, $p \leq 0.003$) and was greatest at lengths 5% longer than optimum fascicle length (mean difference relative to the optimum length: 15 N/mm, $p = 0.008$). The series elastic element stiffness for each participant across the different regions of TA's force-length relation is shown in Fig. 4B. Note that the linear fits of the horizontal fascicle displacement and TA tendon force data between 40% and 90% of maximum force had adjusted $R^2$ values of 0.98 (0.01) with a minimum of 0.89, and root-mean-squared errors (RMSEs) of 6% (2%).

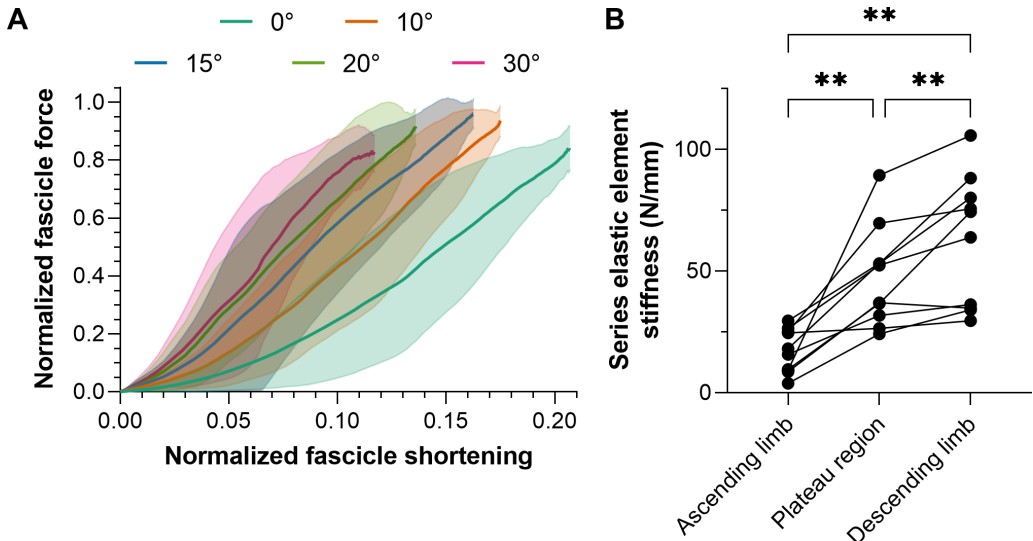

**Figure 4** **Mean (solid lines) normalized tibialis anterior (TA) muscle fascicle shortening magnitudes (A) and TA's mean series elastic element stiffness on different regions of its active force-length relation.** The fascicle force and length at the instant of maximum active force production were used to normalize individual data before the between-subject mean was taken. The shaded areas in A represent standard deviations ($n = 10$). Series elastic element stiffness values in B were calculated as the slope of the relation between horizontal fascicle displacement and TA tendon force between 40% and 90% of maximum force. The mean series elastic element stiffness at lengths: (1) 5% shorter than optimum fascicle length (*i.e.,* ascending limb of the force-length relation); 5% longer than optimum fascicle length (*i.e.,* descending limb), and; between (1) and (2) (*i.e.,* plateau region) was then taken.

## Tibialis anterior force-length relation

The normalized active TA force-length and force-sarcomere length relations are respectively shown in Figs. 3C–3D and descriptive statistics are reported in Table 2. The active force-sarcomere length predictions, which were calculated using active fascicle lengths from the superficial and deep compartments and assuming an optimum sarcomere length of 2.64 μm at optimum fascicle length (Fig. 3D), had RMSEs from the *Walker & Schrodt (1974)* curve of 5% and 7% of normalized force, respectively. The same predictions had identical RMSEs from a theoretical relation based on unstrained actin and myosin filament lengths. However, the predictions were better over the ascending limb relative to the *Walker & Schrodt (1974)* curve than the theoretical curve (RMSE$_{sup.}$: 3% and 6%, respectively; RMSE$_{deep}$: 4% and 7%, respectively), and the reverse was true over the descending limb (RMSE$_{sup.}$: 6% and 2%, respectively; RMSE$_{deep}$: 10% and 7%, respectively).

## DISCUSSION

Experimental estimation of the human skeletal muscle *in vivo* force-length relation is rare, but potentially important for improving the accuracy and performance of the muscle models used in human movement simulations. This is because model outputs are highly sensitive to parameters defining the force-length curve, such as the optimum

muscle fiber length for maximum active isometric force production (*Scovil & Ronsky, 2006*). In this study, the human tibialis anterior (TA) force-length relation was estimated from dorsiflexor torque–angle curves constructed from healthy, young women and men. Maximum active dorsiflexion torque was produced between 0–15° plantar flexion. Maximum active TA fascicle force was produced between 10–15° plantar flexion. Greater variability in active compared with passive fascicle lengths at each plantar flexion angle increased the width of the estimated *in vivo* active force-length relation and reduced the root-mean-squared error relative to the normalized active forces predicted based on a simple scaled sarcomere model. These findings are in line with *in situ* observations (*Moo, Leonard & Herzog, 2020*; *Muhl, 1982*; *Winters et al., 2011*), which also suggest that sarcomere lengths shift leftward from the descending limb of the force-length relation under activation (*Binder-Markey et al., 2023*), and indicate that the human TA operates on the ascending and descending limbs of its force-length relation during maximal voluntary contractions performed between 0° and 30° plantar flexion.

## Dorsiflexor torque–angle relation

Considering that TA's force-length relations were estimated based on dorsiflexor torque–angle relations, the validity of the latter relations will be discussed first. Previous studies have reported mean maximum dorsiflexion torques from 25 to 50 N · m at 10°, 15°, and 20° plantar flexion (*Billot et al., 2011*; *Fukunaga et al., 1996*; *Koh & Herzog, 1995*; *Marsh et al., 1981*; *Van Schaik, Hicks & McCartney, 1994*). These findings largely agree with the results of this study (Fig. 5). The more plantar flexed angles of maximum torque production (25−30°; Fig. 5A) reported by *Van Schaik, Hicks & McCartney (1994)* and *Fukunaga et al. (1996)* could be because the total measured rather than active joint torques were reported. Future investigations should explore if moment arm length differences between women and men help to account for variability in the optimum angles of maximum torque production.

## Tibialis anterior force–angle relation

We found that TA's mean force–angle relation had a clear plateau region between 10° and 15° plantar flexion (Fig. 5B), which agrees with one previous study that electrically stimulated the common peroneal nerve for two seconds at 50 Hz with a supramaximal intensity (*Oda et al., 2005*). Another previous study that electrically stimulated TA's muscle belly for two seconds at 100 Hz with a maximal intensity found a steeper ascending limb (*Maganaris, 2001*), but we believe this was potentially because of increasing plantar flexion torque contributions from the peroneus longus and brevis muscles due to current leakage at more dorsiflexed angles (*Gandevia & McKenzie, 1988*). The relatively more plantar flexed angle of optimum force production in that study (*Maganaris, 2001*) might have also been due to the same phenomenon. However, a future study that non-invasively and intramuscularly stimulates TA's proximal motor point would be required to support our speculation.

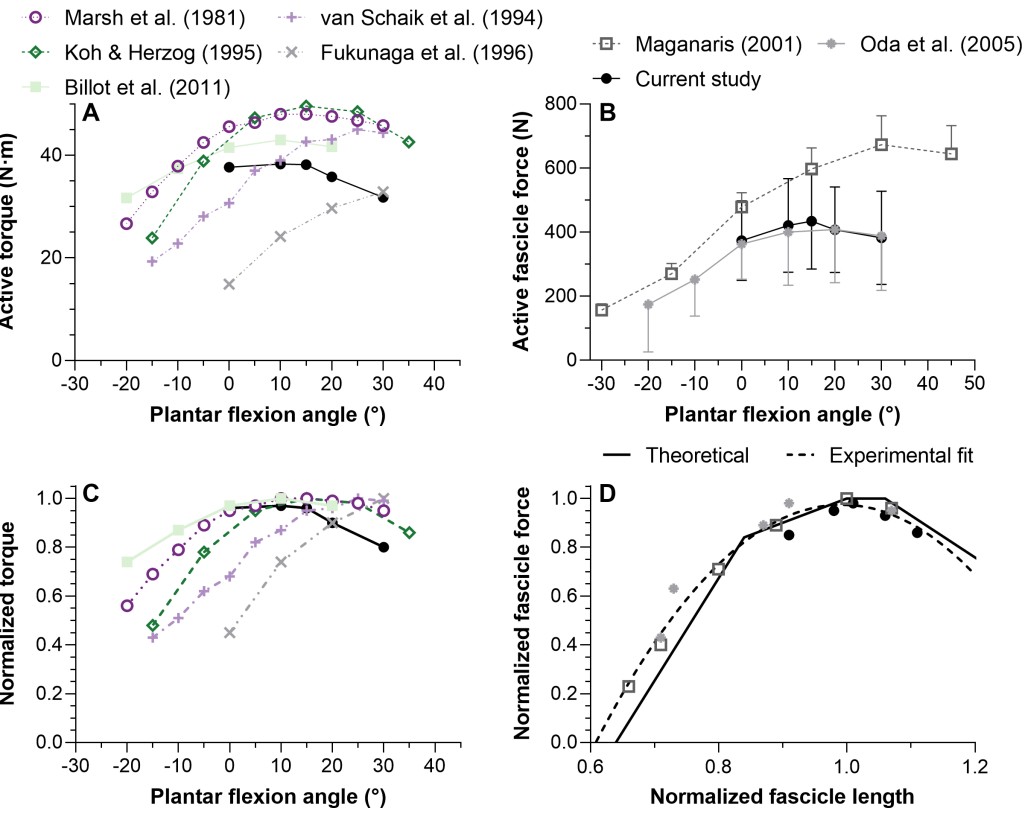

**Figure 5** **Mean maximal active dorsiflexor torque–angle (A) and tibialis anterior active force–angle (B) relations and the normalized torque–angle (C) and theoretical force-length relations (D).** Torque-angle relations were determined during maximal voluntary fixed-end dorsiflexion contractions and come from up to seven studies, including this one, which investigated a combined total of 59 women and 75 men (~19–40 years). Force-angle and force-length relations come from three studies on a combined total of 7 women and 19 men (~23–33 years). Maximum active torque or force and the corresponding ankle angle or fascicle length were used to normalize the mean data. Mean data was either copied from a table or digitized from a figure within the relevant manuscript using WebPlotDigitizer (https://automeris.io/WebPlotDigitizer/). Data from one leg from different sexes and athletic groups were averaged and uncorrected torques were used. The error bars in B reflect standard deviations. The ascending limb of the theoretical force-length curve in D was based on values reported in Fig. 12 of *Gordon, Huxley & Julian (1966)*, whereas the descending limb was based on values reported in Fig. 13B of *Walker & Schrodt (1974)*. The reason for using different regions from different curves is explained in the discussion. The quadratic fit in D resulted in an adjusted R-square of 0.96 and root-mean-squared error of 5%, and was based on the equation: $y(x) = ax^2 + bx + c$ where $x$ and $y$ were the estimated *in vivo* normalized lengths and forces, respectively, and: $a = -6.62505166358072$; $b = 13.1450465981991$; $c = -5.5461683088274$.

## Muscle-tendon unit compliance

Our findings build on one previous investigation (*Maganaris & Baltzopoulos, 1999*) by showing that fascicles from TA's superficial and deep compartments shorten and rotate similar amounts at the same ankle joint angle to produce similar amounts of force and work during maximal voluntary contractions. Based on previous findings (*De Brito Fontana & Herzog, 2016*; *Ichinose et al., 1997*; *Reeves & Narici, 2003*), it is unsurprising that muscle fascicle shortening magnitudes from both compartments increased with

decreasing muscle–tendon unit length, despite lower TA fascicle forces at shorter muscle–tendon unit lengths. This is because the maximum TA tendon stress that has been reported is 25 MPa (*Maganaris & Paul, 1999*), which indicates that the TA tendon operates within the toe-region (*i.e.*, <30 MPa) of its force-length relation (*Ker, Alexander & Bennett, 1988*), even during maximal-voltage percutaneous electrical stimulation. Assuming that mammalian muscle-specific tension is 0.3 MPa (*Jayes & Alexander, 1982*), and TA's physiological cross-sectional area is 22 cm$^2$ based on *in vivo* estimates from four separate studies involving 9 women and 41 men (*Fukunaga et al., 1996*; *Fukunaga et al., 1992*; *Handsfield et al., 2014*; *Maganaris et al., 2001*), the maximum force within TA's tendon during fixed-end contractions should not exceed 660 N. This value falls within the range of active fascicle forces estimated from either muscle compartment in this study of 193–721 N, noting that the error introduced by not accounting for fascicle angle is small (<2%) (*Maganaris & Baltzopoulos, 1999*). The maximum TA tendon stresses calculated from the current data would thus range from 9–35 MPa, based on a mean tendon cross-sectional area of 20.5 mm$^2$, which was measured by *Maganaris & Paul (1999)*. These calculations support their interpretation that the TA tendon operates in its toe region during everyday movement, but add that the TA tendon can operate within its linear region during maximal voluntary contractions in heavier and taller individuals.

Interestingly, even at estimated tendon stresses of 34 MPa in one individual, TA's series elastic element stiffness was ~16 N/mm higher on the descending limb than plateau region of TA's force-length relation. As the tendon should operate within its linear region at stresses over 30 MPa (*Ker, Alexander & Bennett, 1988*), this indicates that a constant relation between fascicle shortening magnitude and active fascicle force might not exist for the TA across muscle–tendon unit lengths. This could be because TA's apparent central aponeurosis stiffness is not just dependent on muscle force (*Scott & Loeb, 1995*), but might also increase with increasing muscle–tendon unit length (*Raiteri, Cresswell & Lichtwark, 2018*). We speculate that transverse curvature of the central aponeurosis contributes differently to its apparent longitudinal stiffness at different muscle–tendon unit lengths because transverse curvature is not fixed; the underlying principle is akin to how differences in transverse curvature along a currency note can promote flexibility or rigidity in the longitudinal direction (*Venkadesan et al., 2020*).

## Tibialis anterior force-length relation

Normalizing individual TA force-length relations by the maximum force and the active fascicle length at maximum force and subsequently taking the between-subject mean resulted in estimates that agree with the normalized human muscle force-sarcomere length curve from *Walker & Schrodt (1974)*. Force normalization was necessary as men were generally heavier and taller, which allowed them to generate higher muscle forces than women. This finding is compatible with the idea that muscle strength is dependent on body size (*Pollock & Shadwick, 1994*) rather than biological sex (*Sepic et al., 1986*). Predicted and normalized forces had lower root-mean-squared errors (5% *versus* 6%) from the *Walker & Schrodt (1974)* relation, which assumes actin and myosin filaments strain during force production (*Wakabayashi et al., 1994*), than a theoretical relation

based on unstrained actin and myosin filament lengths (*Burkholder & Lieber, 2001*). However, this was because of lower errors on the ascending (4% *versus* 7%) rather than descending limb (7% *versus* 4%).

Unfortunately, very few researchers have compared their estimated *in vivo* force-length relations with a theoretical relation, with one *in vivo* study finding a twice as wide human rectus femoris force-length curve than predicted based on a simple scaled sarcomere model (*Herzog & ter Keurs, 1988*). Potential reasons for such a large discrepancy are that fascicle lengths were not measured and vasti muscle force and torque contributions were assumed to be identical at the same knee joint angle, but different hip joint angles. In contrast, one study that measured *in situ* sarcomere lengths (less than one-millionth of all sarcomeres) and forces from the mouse TA found similar experimental and theoretical force-length relations (*Moo, Leonard & Herzog, 2020*). In combination with the current study, these findings suggest that imaging a limited region of the muscle during contraction (independent of the muscle scale) can result in relatively accurate predictions of the normalized forces theoretically predicted based on a single scaled sarcomere.

Although the *Walker & Schrodt (1974)* force-length curve had the least deviation from the *in vivo* estimates of this study, this is not the case when all TA force-length data from the literature are considered (Fig. 5D). The *Walker & Schrodt (1974)* curve predicts no active force production at normalized sarcomere lengths below 48% of optimum length. However, the normalized *in vivo* estimates suggest no active force production at around 60% of optimum length, which agrees better with the normalized data from *Gordon, Huxley & Julian* (*1966*; see their Fig. 12). There are several potential explanations for why active force generation becomes zero below a certain normalized muscle length that go beyond the scope of this study (*e.g.*, the latest classical-physics model indicates that this is due to increases in lattice spacing as sarcomere length decreases and not due to thick filaments colliding with Z discs or double overlap of thin filaments (*Rockenfeller, Günther & Hooper, 2022*)), but from the current data it is implausible that TA fascicles would ever shorten to a length where they produce zero active force.

## Limitations
Several non-trivial assumptions were required to non-invasively estimate force from net joint torque recordings in this study that should be kept in mind. Literature-based TA muscle–tendon moment arms were used to estimate dorsiflexion forces at different joint angles because we did not have access to a MRI scanner and we did not employ the tendon excursion method because it is invalid when passive muscle force contributions change (*Olszewski, Dick & Wakeling, 2015*). Additionally, moment arm length measurements came from men because no systematic investigation has been performed on women. Moment arm lengths were not scaled based on anthropometric measurements from this study because body size and joint size do not appear to be good predictors of moment arm (*Tsaopoulos, Maganaris & Baltzopoulos, 2007*). Negligible co-contraction was assumed based on findings from two previous studies (*Raiteri et al., 2016b*; *Raiteri, Cresswell & Lichtwark, 2015*), and these findings at one ankle joint angle were generalized to all the joint angles tested here. Negligible passive plantar flexor

force contributions at more dorsiflexed angles were also assumed, which could have reduced the measured net ankle joint torques, particularly on the ascending limb of the dorsiflexor torque–angle relation as passive plantar flexor forces can increase when the plantar flexors are slightly stretched during fixed-end dorsiflexion contractions (*Raiteri et al., 2016b*). Fixed dorsiflexor relative force contributions across ankle joint angles were further assumed because no validated method is available to determine individual muscle force contributions to a measured net joint torque, which is a major limitation because TA's force contributions relative to the other dorsiflexors probably change with ankle joint angle. However, the experimental setup was designed to limit toe extensor torque contributions to the measured ankle joint torque.

The level of voluntary activation was also not estimated because non-invasive electrical stimulation can activate the peroneal muscles and cause a superimposed plantar flexion twitch torque (*Gandevia & McKenzie, 1988*) that invalidates the estimation. Changing passive muscle force contributions due to fascicle shortening at longer muscle–tendon unit lengths (*Hessel et al., 2020*) were also not accounted for because of difficulties with estimating TA's passive force-length relation from the passive ankle joint torque–angle relation, which could have resulted in underestimations of active force at more plantar flexed angles. Finally, although participants did not repeat maximal voluntary efforts at each ankle joint angle, repeated maximum voluntary contractions at the optimum angle for torque production had a mean torque difference of 5.3% (5.5%) across all participants, and the first of the repeated trials most frequently (25/31) exhibited the highest peak torque, which gives us confidence that additional contractions would not have resulted in closer to maximal efforts.

## CONCLUSIONS

This study was designed to determine which ankle joint angles and lengths the young adult human tibialis anterior (TA) attains its maximum active isometric force. There was a 10° wider range of optimum angles of torque production (0−15° plantar flexion) than force production (10−15° plantar flexion) in healthy women and men. A quadratic fit of all normalized TA force-length data from the literature (Fig. 5D) agrees well with the ascending and descending limbs of normalized data from *Gordon, Huxley & Julian (1966)* and *Walker & Schrodt (1974)*, respectively, and this fit is provided to inform musculoskeletal simulations investigating TA function. Future modeling studies should consider modifications that allow a muscle–tendon-unit-length-dependent stiffness of the series elastic element because based on data from this study (Fig. 4), assuming a constant stiffness at each ankle joint angle would result in inaccurate contractile element length and force estimates. Future experimental studies are needed to explore whether neural drive is increased at shorter muscle–tendon unit lengths to account for the reduced mechanical output arising from increased fascicle shortening magnitudes and velocities under greater muscle–tendon unit compliance.

## ACKNOWLEDGEMENTS

We thank the participants for their time commitment.

### Funding

This work (RA3308/1-1) was funded by the German Research Foundation (DFG). There was no additional external funding received for this study. The funders had no role in study design, data collection and analysis, decision to publish, or preparation of the manuscript.

### Grant Disclosures

The following grant information was disclosed by the authors:
German Research Foundation (DFG): RA3308/1-1.

### Competing Interests

The authors declare there are no competing interests.

### Author Contributions

- Brent J. Raiteri conceived and designed the experiments, performed the experiments, analyzed the data, prepared figures and/or tables, authored or reviewed drafts of the article, and approved the final draft.
- Leon Lauret performed the experiments, analyzed the data, prepared figures and/or tables, and approved the final draft.
- Daniel Hahn conceived and designed the experiments, authored or reviewed drafts of the article, and approved the final draft.

### Human Ethics

The following information was supplied relating to ethical approvals (*i.e.*, approving body and any reference numbers):

The experimental protocol was approved by the Ethics Committee of the Faculty of Sport Science at Ruhr University Bochum (EKS V 33/2019).

### Data Availability

The data is available at Zenodo:

Raiteri, Brent James. (2022). Force-length data (Version 1) [Data set]. Zenodo. https://doi.org/10.5281/zenodo.7421470

The code is available at Zenodo:

Raiteri, Brent James. (2022). Force_length_analysis (Version 1). Zenodo. https://doi.org/10.5281/zenodo.7411400

Raiteri, Brent James. (2022). UltraTrack_v5 (2&3). Zenodo. https://doi.org/10.5281/zenodo.7411280

## Supplemental Information

Supplemental information for this article can be found online at http://dx.doi.org/10.7717/peerj.15693#supplemental-information.

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
