# Peer review of "The force-length relation of the young adult human tibialis anterior"

_PeerJ, doi:10.7717/peerj.15693_

## Round 0.1 · original submission · Major Revisions

While the work appears to be executed well, reviewers 2 and 3 both wanted to see a better presentation of the novelty, the rationale, and a clear scientific question or hypothesis. Both commented that the hypothesis seemed artificial which requires a more clear presentation of the rationale for the study. We are looking forward to your revision to address the reviewer comments.

·

Basic reporting

Please see report at end.

Experimental design

Please see report at end.

Validity of the findings

Please see report at end.

Additional comments

The submitted paper by Raiteri et al. investigates a basic property of muscle – the force-length relationship. Through the use of muscle ultrasound, careful dynamometry measurements, and published moment arms of the dorsiflexors, the authors construct a force-length relationship of the tibialis anterior (TA). This is not a trivial feat, as moving from joint level torque angle relationships to a muscle’s force-length relationship is very very challenging and not without many experimental considerations to ensure proper estimations – kudos to the authors on this challenging work! Importantly, the authors include both males and females in this data set, which should help to improve the generalizability of the findings across the sexes. These data are extremely important, and will serve to improve available data sets for those individuals wanting to model in vivo human muscle function. The paper is well-written, and the authors acknowledge key limitations of the work, I offer a few comments/suggestions/questions below to help improve clarity and identify any areas the authors may wish to reflect or expand upon.

Line 19: Consider rewording to “…knowledge of the muscle’s length at which maximal active isometric force is attained…”

Line 34: Not clear what muscle compartments mean here in the abstract, were multiple areas imaged?

INTRODUCTION: Very clear identification of the knowledge gap, nice justification and background on the proposed muscle to use to address the gap, and a clear purpose.

Line 58 – is the lack of data on females ‘surprising’? or understandable given the lack of studies including males and females?

Line 65 - remove ‘an’

Line 92-94, in the hypothesis statement, could similar FL and SSN across the sexes be supported with refs/data from the literature? Or expanded upon. If the F-L relationship is not normalized to Fl then differing SSN based on differences in FL would influence the curve, no?

Line 94: Even if there are a similar # of serial sarcomeres, would any possible differences in tendon compliance between males and females affect the MTU length at a given joint angle? I am somewhat familiar with work from Dr. Jakobi (PMID: 29049892, DOI: 10.1139/apnm-2017-0289 ) on sex-differences and in series compliance. If the authors think this is something worth considering in the introduction to help set up the hypothesis, please feel free to expand on this idea.

METHODS: Nicely detailed, clear to follow.

Please state training (or lack there of) background of participants.

Line 121: Consider a comment on consistent pressure of the ultrasound probe?

Line 124: Some readers may benefit from more information on the compartments of the TA or additional refs here to point them in the right direction.

Lines 147-151: Is this some sort of a warm up/pre-conditioning procedure? Was there a familiarization session? How much rest was given before data were collected? How did the authors ensure the contractions were maximal? Would voluntary activation (ability to generate a maximal effort) differ across joint angles? I do think the DFers are easy to activate near maximally with little to no practice (I see this with the interpolated twitch technique >95%) as compared with muscle groups like the PFers (usually <90% on first attempt). Without multiple contractions at each joint angle, how confident are you the efforts are indeed maximal? As well, EMG was mentioned (Line 130, and abstract) but does not appear in the results. Was antagonist activation measured/quantified across joint angle? This may also be considered in the limitations section as joint-angle dependent changes in antagonist activation could affect the DFer torque angle relationship.

Line 193: Do these reported moments arms account for differences across the sexes? Is there any reason to even expect a difference? Please clarify.

Line 197 – A technical note here: I would urge the authors to think carefully about the 50% value that is assumed to reflect force contributions of the TA to total DFer force. In those original papers that measured torque/force, did the authors have the ‘toes’ secured or free to move?? That is one factor that affects the relative contributions of the ext hallucis longus and ext digitorum longus to DF force. The authors use a set up which does not cover the toes (good!), thus, I wonder if their subjects were closer to 60+% force contribution from the TA. I do not expect any changes to the analysis, just something for the authors to consider, and possibly a point in the discussion if they think that is warranted.

RESULTS:

Some exemplar ultrasound images would be nice to see included, this will also help with interpreting the methods/set up.

Figures - Akagi et al. 2020 also have torque angle relationships of the dorsiflexors if the authors wanted to include these data for completeness https://doi.org/10.1152/japplphysiol.00280.2020 here and the discussion.

Tables 1 & 2 & 3 – Please note n=? males/females for each joint angle. Justification of differing participants per joint angle would be helpful. Given the effect of sex on joint angle, I think separating the data into male and female offer more clarity to readers interpreting the data? For example, Did males and females differ in ‘fascicle shortening amplitudes’ – this info may be helpful in interpreting the results.

DISCUSSION: Positions the findings in the context of the literature, and acknowledges important assumptions/limitations of the work, well-done.

Line 330 – Did you assess ROM? If not do you have any refs to support a larger ROM in females.

Reviewed by Geoff Power.

Reviewer 2 ·

Basic reporting

**Clear, unambiguous, professional English language used throughout.**

I am not a native speaker, so I will not judge the language correctness. But I can clearly understand the content of the paper, throughout, if not mentioned otherwise.

**Intro & background to show context. Literature well referenced & relevant.**

The introduction does not clearly state the gap in the research and the novelty over previous work. A first quick look into the literature revealed the paper

https://onlinelibrary.wiley.com/doi/abs/10.1046/j.1365-201x.2001.00799.x

which is also cited in the paper. I don't have access to it so for me it is not really possible to judge the novelty of the presented work.

Furthermore, the hypotheses come a bit out of the blue and are not motivated. It sounds like you looked at the data and then derived hypotheses from there. At least I do not understand the derivation of your hypotheses from the rest of the introduction.

It was hypothesiszed... Did you hypothesize or was this a hypothesis that you got from the literature?



Also line 42: There are a couple of experiments that measured muscle force in vivo, e.g.,

https://www.sciencedirect.com/science/article/abs/pii/S0268003398000321



I would suggest to

1: clearly state: the open question/purpose. Is it the force-length relation of female subjects? This is unclear.

2: clearly state: the novelty of this paper is... (did you adopt previous methods and then applied them to females or did you also improve the methods. This is unclear.)

3: clearly stat: the relevance -> for modelling.

All of this in the last paragraph of the intro.



**Structure conforms to PeerJ standards, discipline norm, or improved for clarity.**

yes

**Figures are relevant, high quality, well labelled & described.**

yes

**Raw data supplied (see PeerJ policy).**

yes

Experimental design

**Original primary research within Scope of the journal.**

yes

**Research question well defined, relevant & meaningful. It is stated how the research fills an identified knowledge gap.**

See comments regarding introduction above.



**Rigorous investigation performed to a high technical & ethical standard.**

The approach relies on maximum voluntary contractions. In my understanding, the better strategy to egt systematic data on maximum force is an on-top stimulation, as e.g., described in

Mau-Moeller, A., Bruhn, S., Bader, R., and Behrens, M. (2014). The relationship between lean mass and contractile properties of the quadriceps in elderly and young adults. *Gerontology* 61, 1–5. doi: 10.1159/0003 68656

This limits the study's validity.

Fatigue was not measured, but could influence the results.



**Methods described with sufficient detail & information to replicate.**

The muscle-tendon moment arm length model is not described and I could not find it in the given reference (Maganaris et al., 1999).

Validity of the findings

**Impact and novelty not assessed. Meaningful replication encouraged where rationale & benefit to literature is clearly stated.**

See comments related to the introduction

**All underlying data have been provided; they are robust, statistically sound, & controlled.**

Data is provided and statistically plausible. I am not convinced that it is controlled enough to make claims towards improved accuracy of the parameters for modelling. See other comments.

**Conclusions are well stated, linked to original research question & limited to supporting results.**

The novelty remains unclear, see other comments

Additional comments

The authors report on an experimental study with the aim to measure muscle parameters relevant to muscle modelling. From what I understood, the main novelty of the data set is the recruitment of male and female subjects and the comparison between the two groups. Accurate measurements of muscle parameters is highly relevant for accurate predictions of musculo-skeletal models and therefore the aim of the study is relevant and could be an important contribution to the community.

Unfortunately, I have concerns about the validity of the approach and I am unsure with respect to the novelty/originality of the study.

My major concern is: ultimately, the authors claim that their parameters could be used for better force predictions. Yet, they fail to show this. The study would greatly benefit from implementing their parameters in musculo-skeletal models and showing the improved prediction accuracy. Otherwise, the made assumptions about a general moment arm relation (which is not even given), as well as the complete muscle force recruitment by voluntary contraction and the distribution of muscle force between synergetic muscles, may just hinder the determination of real parameters. Last but not least, the conclusion about the difference between male and female subjects also relies on the same lever arm equation, as far as I understood. While these points are mostly addressed in the limitations section, ultimately, the benefit and novelty of this study remains unclear to me.

For more details, please see my other comments.

·

Basic reporting

This descriptive study contributes to our understanding of the force-length relationship of tibialis anterior muscle. The study is well written, and the rationale is presented in logical way with sufficient field background and appropriate reference to the literature. Having said that, the hypotheses (based on the number of sarcomeres between men and women and differences across compartments) seem rather artificial and the rationale behind is not presented . The structure of the paper is logical and clear, with carefully prepared Figures.

Experimental design

The study is descriptive in nature and was aimed at estimating TA’s force-length relation for men and women from dorsiflexor torque-angle curves from 16 subjects. The research questions stated at the end of the introduction (hypotheses) were:
- Are the angles of maximum active dorsiflexion torque and maximum active TA fascicle force similar between women and men?
- Are TAs muscle fascicle shortening amplitudes during fixed-end contractions larger at shorter than long muscle-tendon unit lengths?
- Are TAs muscle fascicle shortening amplitudes different between muscle compartments?
The three research questions however do not fit easily to the introduction and its meaningfulness and relevance are not currently satisfactorily explored in the introduction (sex and compartment effect).
The study involved 16 subjects (8 male and 8 female) and therefore is limitated in sample size to address the effect of sex. Authors wrote a detailed description of the methods but some parts could be revised to improve clarity and allow replication.
The fact that some subjects produced contractions only at one angle (have I misunderstandood this (?) – please improved clarity) challenges the validity of the conclusions.

Validity of the findings

The study estimates muscle force from joint torque and fascicle length from one plane ultrasound measurements. The procedures can be considered standard and are generally accepted in the literature. However, there are known limitations that need to be carefully described. The authors do a good job in discussing some of these limitations in the discussion but I believe other important limitations were not mentioned. The conclusion that the TA works in the ascending and descending regions of the force-relationship seems questionable as the largest differences between angles (~7%) seems to be smaller than the typical error for torque measurements. Perhaps a more accurate conclusion is that TA works around the plateau of the force-length relationship. Additionally, the study rationale is built around the estimation of muscles forces using musculoskeletal models and our current knowledge regarding TA’s properties. However, no dynamic contractions were performed, and the TA force-velocity relationship (that highly influence force production) is not mentioned or discussed.

Additional comments

This study aimed to estimate TA’s force-length relation for humans from dorsiflexor torque-angle curves of 16 subjects.

Specific comments:


“angle-specific tendon moment arm lengths while assuming a fixed 50% force contribution of TA to the total dorsifleexor force and accounting for fascicle
angles”

- Why was a fixed (across joint angles) 50% force contribution considered? Please discuss the potential limitations of this assumption.
- How was fascicle angle accounted for? Please discuss the potential limitations of this procedure in providing an estimate of fascicle force.

“Maximum torque was most frequently produced at 5° plantar flexion by women and 15° plantar flexion by men (p = 0.024). However, maximum TA fascicle force was developed most frequently at 15° plantar flexion by both sexes (p = 0.689).”

- Was this sex specific shift (from 5 to 15 degrees) due to differences in moment arm or differences in fascicle angle?

Line 62-63: “Moreover, these muscle-tendon units have unique moment arms that are affected by the joint.s configuration (Murray et al., 2002).”
– and by force production. Please discuss.

Line 83-85: “TA contributes 45-52% of the maximal voluntary dorsiflexion torque as assessed via electrical stimulation of its muscle belly (De Zee and Voigt, 2002; Maganaris and Paul, 2000) and based on its relative physiological cross-sectional area (Brand et al., 1986).”

- Phsiological cross sectional area can be a poor predictor of force distribution (i.e. doi: 10.1242/jeb.188292 and 10.1016/j.jbiomech.2018.07.007

Line 87: “TA fascicle length changes and sarcomere length changes are significantly and positively correlated.”
- More importantly than saying they are related is to inform the strength of this correlation.


Line 91. “It was hypothesized that the angles of maximum active dorsiflexion torque and maximum active TA fascicle force would be similar between women and men because of a similar number of sarcomeres in series for a given fascicle length.”
- It does not seem appropriate to build your hypothesis based on a second hypothesis (sarcomere number) that will not be tested.
Why did the authors hypothesize that “Normalized force-length relations
based on active and passive fascicle lengths were predicted to poorly match theoretical estimates based on human actin and myosin filament lengths.” Also, please correct the verb tense as it currently refers to the past. The rationale for this hypothesis was not presented.

Line 143: “The footplate was inverted by 5° about its longitudinal axis to facilitate alignment.” Why?


Line 149 “then performed at least one maximal.”
- Not clear

Line 157: “Contractions were not repeated at all tested ankle angles”
- Not clear

Line 200 to 205:
- It is not clear how the measurements of work and slack joint angle fir to the research questions. I feel the manuscript coherence (objective-methods- results) could be revised and improved.

Line 213: “not be synchronized with their analog data to a time difference of within one frame”
- Not clear

Line 227 and 228.
- What is the rationale for this inclusion criteria? It seems strange that you calculate ICC only for subjects that presented a certain (desirable) level of reproducibility. Additionaly, why was ICC calculated for work and not force?

Results:
Line 239. “Maximum dorsiflexion torque occurred between ankle angles from -5° to 15° plantar flexion.”
- How confident are the authors in this result. It seems that there was only one angle for which data from the 16 subjects was available. How many subjects cleary presented a peak (i.e. force decreasing for the neighbouring angles? I also do not understand why the number of participants is different between table 1 and 2.

Line 241: “The maximum difference in mean torque between -5° to 20° plantar flexion was 2.7 Nm (7%),”
- Again, if the difference across angles is that small (smaller than the typical MDC for torque measurements), I am not sure the conclusion that peak torques occur at -5 (e.g) is strong. Could we conclude that the TA works across the plateau and with no clear peak across angles?


Line 250: “n1 = n2 =8)”
- What is this?

Line 257: “The maximum difference in mean force between 5 and 20° plantar flexion was only 21 N (5%), and there was a maximum difference in mean force of 105 N (26%) between -5° and 15° plantar flexion”

- It is not clear to me why these results are shown. Could you potentially rephase to improve clarity?

Line 272 “-9° (6°)”
- Are these mean and SD. If so, please be consistent with format.

Line 272 “and it was based on the instant of fascicle lengthening from either the superficial or deep muscle compartment.”
- Not clear
-
Line 296 -300.
- Given that this study was aimed at testing for differences between men and women, why are the anthropometric differences treated as exploratory. I would suggest authors to build the rationale for this analysis in the introduction if the hypothesis regarding the effect of sex is kept.

A vast part of the discussion is associated to the research question (ii) regarding the dependence of fascicle shortening on joint angle (activation is also mentioned in future work). Please consider citing the work doi:10.1007/s00421-016-3381-3 which clearly showed the dependence of fascicle shortening on joint angle across activation levels.


In the limitations, please consider discussing:
- Changes in moment arm with force production
- Interactions between TA and its surroundings (intermuscular pressure, epimuscular force transmission, etc ) and how that might affect force generation and more importantly the shape of the force-lenth curve.
- The sample size.


Figures: Instead of stating “sample sizes are provided in Table 2”, please indicate in the Figure legend. * I see now that sample size is not the same across angles. I consider that a problem as deviations in the shape of the force-length curve occurs across subjects and the fact that the effect of angle is not entirely based on repeated measurements is an additional source of bias.



Figure 2
“Incorrect sarcomere length predictions were calculated by dividing passive fascicle lengths at each tested ankle angle by the quotient of passive fascicle length at the angle of maximum force and 2.64 μm.”

- Do you mean the quotient between fascicle length and the estimated number of sarcomeres? Please revise.
- Please specifically refer to Figure B and C in the legend.
- In D, it is not clear what you mean by passive. From the legend it is possible to understand what “incorrect passive” means but not the passive data

Figure 5
“The ascending limb of the theoretical forcelength curve in D was based on values reported in Fig. 12 of Gordon et al. (1966), whereas the descending limb was based on values reported in Fig. 13B of Walker and Schrodt (1974).”

- Why did authors used different references for the different regions of the force-length relationship?


Thank you for the opportunity to review your paper. I look forward to reading a revised version.

---

## Round 0.2 · Minor Revisions

Thank you for the revision which was overall quite responsive to the reviewer comments.

Please submit a final revision that addresses the remaining comments of Reviewer 2. If the resubmission looks good to me, I will accept the manuscript without sending it back to the reviewers.

·

Basic reporting

The authors addressed my initial concerns.

Experimental design

The authors addressed my initial concerns.

Validity of the findings

The authors addressed my initial concerns.

Additional comments

The authors addressed my initial concerns.

Reviewer 2 ·

Basic reporting

A recent paper that could be interesting and worth citing in this context: https://physoc.onlinelibrary.wiley.com/doi/full/10.1113/JP284092

Experimental design

No comment

Validity of the findings

Novelty is now clear. See below.

Additional comments

I would like to thank the authors for their thorough response and revision.

The introduction and therefore the purpose and novelty, as well as the presentation of the results of the study are much clearer now. Also the methods section regarding the series elastic stiffness has improved. I can now follow the arguments and get the point of the paper. Indeed, I recommend publication and only propose to consider a a few small comments/suggestions:


Line 101: Maybe add one sentence to motivate this hypothesis. Did you expect this due to the force-length relation only or due to the lever arm and why would you expect this optimum to be in the range of joint limits and is this the same angle for dorsiflexion torque and fascicle force production or do they not need to coincide?

Line 217 and 225: I personally do not like reading equations in text form. Have you ever tried to read Newton's original texts? It is much easier to understand in the current text book form F=ma. To me an equation is much more intuitive to read and understand and I would recommend to note the calculation of central quantities of your study down as equations.

Line 410: is there any hypothesis or speculation about a mechanisms that could explain the change in stiffness with muscle-tendon unit length?

---

## Round 0.3 · accepted · Accept

Thank you for your final revisions. The manuscript is now acceptable for publication.